# Generating Cerebral Vessel Trees of Acute Ischemic Stroke Patients using Conditional Set-Diffusion

**Thijs P. Kuipers**[1,2]                                      T.P.KUIPERS@AMSTERDAMUMC.NL
[1] *Department of Biomedical Engineering and Physics, Amsterdam UMC, The Netherlands*
[2] *Department of Radiology and Nuclear Medicine, Amsterdam UMC, The Netherlands*

**Praneeta R. Konduri**[1,2]                              P.R.KONDURI@AMSTERDAMUMC.NL
**Charles B. L. Majoie**[2]                                    C.B.MAJOIE@AMSTERDAMUMC.NL
**Henk A. Marquering**[1,2]                           H.A.MARQUERING@AMSTERDAMUMC.NL
**Erik J. Bekkers**[3]                                                       E.J.BEKKERS@UVA.NL
[3] *Informatics Institute, University of Amsterdam, The Netherlands*

**Editors:** Accepted for publication at MIDL 2024

## Abstract

The advancements in computational modeling and simulations have facilitated the emergence of in-silico clinical trials (ISCTs). ISCTs are valuable in developing and evaluating novel treatments targeting acute ischemic stroke (AIS), a prominent contributor to both mortality and disability rates. However, obtaining large populations of accurate anatomical structures that are required as input to ISCTs is labor-intensive and time-consuming. In this work, we propose and evaluate diffusion-based generative modeling and set transformers to generate a population of synthetic intracranial vessel tree centerlines with associated radii and vessel types. We condition our model on the presence of an occlusion in the middle cerebral artery, a frequently occurring occlusion location in AIS patients. Our analysis of generated synthetic populations shows that our model accurately produces diverse and realistic cerebral vessel trees that represent the geometric characteristics of the real population.

**Keywords:** deep learning, diffusion, conditional, cerebral vessels, arteries, stroke, in-silico modeling

## 1. Introduction

Advancements in computational modeling and simulations have enabled the development of in-silico clinical trials (ISCTs). Combined with machine learning methods, ISCTs can provide valuable insights into new medical treatments and devices. ISCTs allow for a reduction of resources spent on developing new devices and treatments by simulating their performance on virtual patient populations, reserving actual clinical trials only for the most encouraging candidates (Konduri et al., 2020; Miller et al., 2023; Viceconti et al., 2016).

ISCTs can aid in the development of novel treatments for acute ischemic stroke (AIS), one of the leading causes of mortality and disability (Phipps and Cronin, 2020). AIS occurs when a thrombotic occlusion reduces the blood flow in a cerebral vessel. Treatment of large vessel occlusions, for example, in the interior cerebral artery (ICA), the middle cerebral artery (MCA) M1 and M2 segments, or the anterior cerebral artery (ACA), involves mechanical thrombectomy, for which in-silico treatment models have been developed and

validated (Luraghi et al., 2021). The training of in-silico treatment models benefits from large populations of patient-specific information derived from radiological images, including segmentations of cerebral vessels and characteristics of the occluding thrombus. Such data enhances the generalizability and robustness of the trials. However, obtaining this data is often expensive and time-consuming. With the advances in generative modeling, synthetic patient populations that can be conditioned on certain auxiliary variables of interest are becoming increasingly popular. Conditioning provides several advantages, such as aligning synthetic data to real-world patient data or customization for specific patient groups.

Various methods have been introduced for generating vessel geometry. Bridio et al. (2023) uses statistical shape modeling (SSM) to generate cerebral vessel anatomies described by their centerlines and diameters. However, SSM approaches are linear models, limiting their ability to model complex and large variations in vessel geometry and topology (Kalaie et al., 2023). Furthermore, conditioning SSM approaches on auxiliary variables is not straightforward (Bannister et al., 2022). Danu et al. (2019) use deep generative voxel-based models for generating vessels. However, voxel volumes have limited resolution due to their high computational cost, making it difficult to convert them to a mesh that is suitable for ISCTs. Wolterink et al. (2018), instead, uses a conditional generative adversarial network (GAN) (Goodfellow et al., 2020) vessel centerlines, which is not limited by resolution. Nevertheless, it assumes a fixed topology, i.e., centerline points have a fixed ordering, and therefore cannot generate structures with bifurcations.

In this work, we present a deep generative diffusion-based method for generating vessel geometry. Diffusion (Ho et al., 2020) has emerged as one of the most powerful generative methods. We address the limitations of voxel-based methods and fixed topology by generating vessel tree centerlines with arbitrary topologies using a conditional set transformer-based architecture (Lee et al., 2019). By representing vessel trees as point clouds, or equivalently, fully connected graphs, our model is not constrained by topology. Since we are interested in computational simulation of stroke treatment for large vessel occlusions, our approach generates vessel tree centerlines with corresponding radii and associated ICA, ACA, M1, and M2 vessel types. However, since our method supports arbitrary toplogy, it is not limited to these specific vessels. The model is conditioned on the presence of an M1 occlusion. We evaluate the synthetic vessels by comparing their geometric characteristics with those observed in clinical data. However, since set-diffusion output is unordered, we present a post-processing centerline-sequencing algorithm for ordering the generated centerlines.

We summarize our contributions as follows. We use diffusion with a set-transformer architecture to generate topology free conditional vessel tree centerlines. A simple post-processing step is used to sequence the unordered centerlines for use in further downstream tasks. Our evaluation shows that our method can generate realistic and unique vessel trees.

## 2. Method

Section 2.1 describes the used dataset and how we represent the centerlines for training the diffusion model. The diffusion process and its set-transformer backbone is described in 2.2. We present the sequencing algorithm that orders the synthetic centerlines sampled from the diffusion model in Section 2.3. An overview of the entire generative process is presented in Figure 1. We make our code publicly available here.

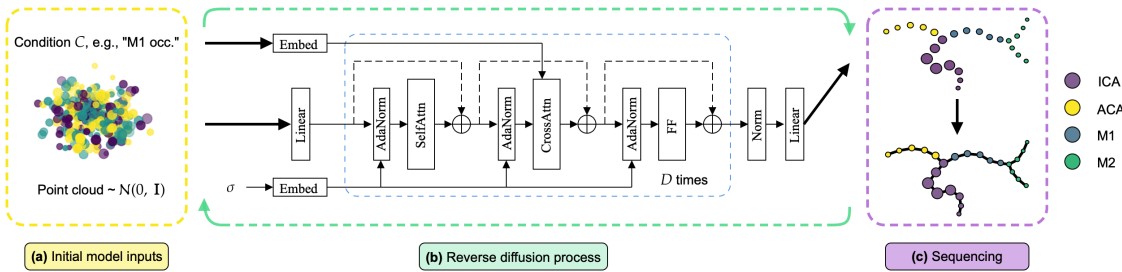

Figure 1: Overview of the generative process. **(a)** Noisy point cloud is sampled and a condition label is provided. **(b)** The noisy point cloud is iteratively denoised during the reverse diffusion process to form a vessel centerline tree. **(c)** The unordered points are sequenced to form the final centerline vessel tree.

### 2.1. Data

In this work, we used data from the MR CLEAN Registry, an ongoing, prospective, observational, multicenter study from 16 EVT-capable hospitals in the Netherlands. The dataset consists of 110 patients with an occlusion in the M1 artery, in either the left or right side of the brain, and sufficient image and segmentation quality. We segmented the intracranial arteries using a vessel segmentation algorithm developed within StrokeViewer (NICO.LAB, Amsterdam). The centerline and geometry characteristics (radius, tortuosity, and bifurcation angle) extraction and artery segment labeling were performed with a semi-automated software: iCAFE(© 2016-2018 University of Washington. Used with permission) (Chen et al., 2018). The vessel trees include ICA, ACA, and the MCA (M1 and M1 segments) on the contralateral hemisphere (without occlusion). To increase the number of data samples, we mirrored vessel trees with occlusion on the left side to the right side. Hence, our dataset consists of 181 training samples (95 with and 86 without an M1 occlusion) and 39 testing samples (20 with and 19 without occlusion).

The vessel trees are parameterized by their centerline, i.e., a set of points $\mathbf{x}_i = (\mathbf{c}_i, \mathbf{h}_i)$ where $\mathbf{c}_i$ contain the point coordinates and $\mathbf{h}_i$ the point features, such as the radius and vessel type. The vessel type is a categorical feature represented as a one-hot encoding. Similar to Hoogeboom et al. (2022), we multiply the categorical features by 0.1 to stimulate the denoising process to first emphasize the shape of the centerline before segmenting it. Note that in practice, $\mathbf{x}_i$ is the concatenation of $\mathbf{c}_i, \mathbf{h}_i$. We sample 256 equidistantly spaced centerline points using linear interpolation that are scaled down by a factor of 24 to be approximately within the range of $[-1, 1]$ and have a standard deviation of 0.5, as this is expected by the EDM (Karras et al., 2022) diffusion formulation.

### 2.2. Conditional Set-Diffusion

The diffusion model consists of three parts. The forward diffusion process adds noise to the input. A denoising function aims to remove the added noise to reconstruct the original input. The reverse diffusion process synthesizes a vessel tree by iteratively denoising noise

from the unit Gaussian distribution. The reverse diffusion process requires 18 steps to synthesize a centerline vessel tree.

**Diffusion Process** We use the EDM formulations introduced by Karras et al. (2022). EDM drastically simplifies the forward diffusion process and the training of the denoising function. Sampling from the diffusion model is also significantly faster, requiring only 18 steps compared to the hundreds of steps such as in (Hoogeboom et al., 2022).

**Denoising Objective** Given a centerline point $\mathbf{x}_i = (\mathbf{c}_i, \mathbf{h}_i)$ and noise $\mathbf{n}_i$, the objective of the denoising function is to map the diffused input back to the original input. The amount of noise $\mathbf{n}_i$ added to the input is determined by the noise level $\sigma$. As $\sigma$ increases, the noisy input increasingly resembles unit Gaussian noise. Formally, the denoising objective is

$$\mathbb{E}_{\mathbf{n}_i \sim \mathcal{N}(\mathbf{0}, \sigma^2 \mathbf{I})} \left[ \frac{1}{N} \sum_{i}^{N} (F_\theta(\mathbf{x}_i + \mathbf{n}_i, \sigma, C) - \mathbf{x}_i)^2 \right]. \tag{1}$$

Here, $F_\theta$ is the denoising function with learnable parameters $\theta$. During training, $F_\theta$ is conditioned on the noise level $\sigma$ and any optional auxiliary conditional information $C$. We use learnable embedding vectors for conditioning on the presence of an M1-occlusion and parameterize. The denoising function $F_\theta$ is modeled by a set-transformer.

**Cross Attention** Our denoising network consists of a series of cross-attention blocks. The attention mechanism allows elements in the input to pass information to each other while being permutation equivariant (Vaswani et al., 2017). Given matrices $\mathbf{X} \in \mathbb{R}^{N \times L}$ and $\mathbf{Y} \in \mathbb{R}^{M \times P}$, with rows denoting individual set elements, we formulate cross-attention as

$$\text{CrossAttn}(\mathbf{X}, \mathbf{Y}) = \mathbf{A}(\mathbf{Y}\mathbf{W}_V) \tag{2}$$

$$\mathbf{A} = \text{Softmax}\left( \frac{\mathbf{X}\mathbf{W}_Q(\mathbf{Y}\mathbf{W}_K)^T}{\sqrt{d}} \right), \tag{3}$$

where $\mathbf{W}_Q \in \mathbb{R}^{L \times d}$ and $\mathbf{W}_K, \mathbf{W}_V \in \mathbb{R}^{P \times d}$ are learnable parameters mapping $\mathbf{X}$ and $\mathbf{Y}$ to sets of queries, keys, and values respectively. In the case where $\mathbf{X} = \mathbf{Y}$, Equation (2) becomes self-attention.

**Set Transformer** The set transformer consists of a series of cross-attention blocks. Each cross-attention block consists of three components. First, self-attention is applied where centerline elements exchange information with each other. Next, conditional information is incorporated via cross-attention, serving as an effective conditioning mechanism (Rombach et al., 2022). In the case of unconditional generation, the cross-attention layer becomes a self-attention layer. Finally, an inverse-bottleneck feed-forward network performs channel mixing. Adaptive layer normalization is applied before each component to inject the noise levels. Pre-normalization in the transformer architecture improves gradient stability, reducing training time and the need for hyperparameter tuning (Xiong et al., 2020).

## 2.3. Unordered Centerline Sequencing

Our generative model generates an unordered set of centerline points. We turn the unordered sets into ordered and cleaned-up connected centerline segments with the following three-stage post-processing algorithm.

**(1) Noise Reduction**   Noise mainly occurs if points are far away from the centerline or if points form clusters. Points that have a nearest-neighbor distance (nn-distance) larger than four times the average nn-distance are removed. Clusters, generally occurring at bifurcations, are reduced by applying furthest-point sampling.

**(2) Sequencing**   Sequencing starts with an empty sequence $s$ to which points are added to the end. We define the last point added to the sequence as the endpoint and the remaining points as candidate points. The candidate point with the minimum distance to the endpoint, weighted by the current direction of the sequence is chosen as the next endpoint. Given the current endpoint and direction $\mathbf{x}_{\mathrm{cur}}$ and $\mathbf{d}_{\mathrm{cur}}$, and a candidate point $\mathbf{x}_i$ with direction $\mathbf{d}_i$, the direction-weighted distance d is calculated as

$$\mathrm{d}(\mathbf{x}_{\mathrm{cur}}, \mathbf{x}_i) = (1 + \alpha \mathbf{d}_{\mathrm{cur}}^T \mathbf{d}_i)\|\mathbf{x}_i - \mathbf{x}_{\mathrm{cur}}\|, \tag{4}$$

where $\alpha$ determines the importance of the current direction. The directions are calculated as $\mathbf{d}_{\mathrm{cur}} = (\mathbf{x}_{\mathrm{prior}} - \mathbf{x}_{\mathrm{cur}})/\|\mathbf{x}_{\mathrm{prior}} - \mathbf{x}_{\mathrm{cur}}\|$ and $\mathbf{d}_i = (\mathbf{x}_i - \mathbf{x}_{\mathrm{cur}})/\|\mathbf{x}_i - \mathbf{x}_{\mathrm{cur}}\|$, where $\mathbf{x}_{\mathrm{prior}}$ is the point prior to $\mathbf{x}_{\mathrm{cur}}$ in the sequence. The point chosen as the initial endpoint is given as the point $\mathbf{x}$ that maximizes the average pairwise inner product between the directions from $\mathbf{x}$ to its $k$ nearest neighbors. Intuitively, the point where its nearest neighbors have similar directions is likely one of the end points of the sequence.

**(3) Segment Merging**   Individual vessel segments are merged by calculating a common bifurcation point. Given an endpoint $\mathbf{x}$ belonging to segment $s$, its nearest neighbors from the remaining segments are candidate bifurcation points. Candidate points with distances greater than 4 times the average nn-distance of $s$ are discarded. The common bifurcation point is the average of the remaining candidate points.

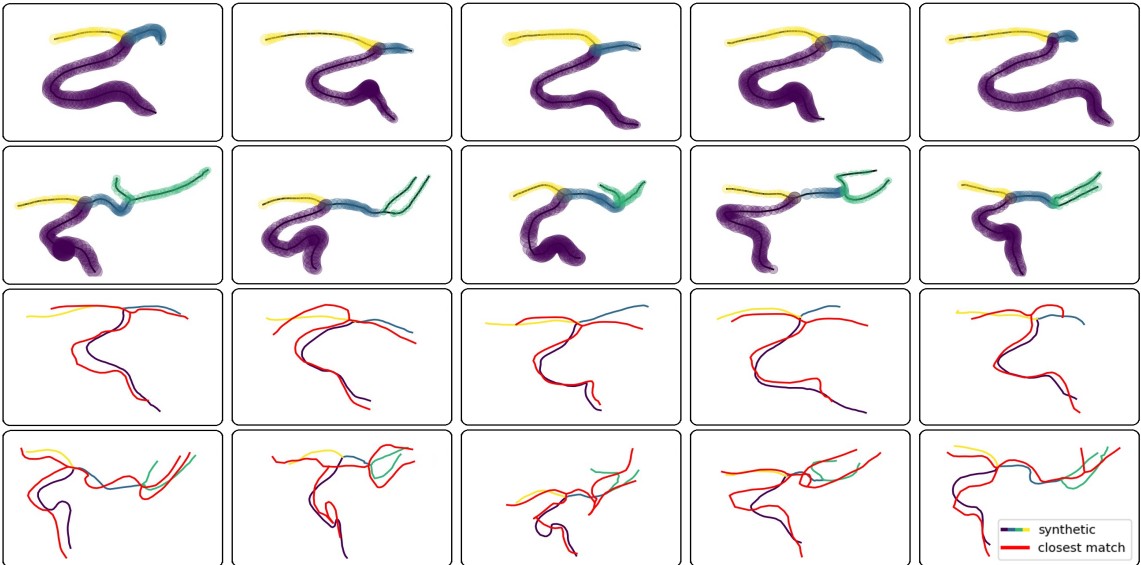

Figure 2: Synthetic vessel trees. The bottom two rows display the skeletons with the closest match from the training set.

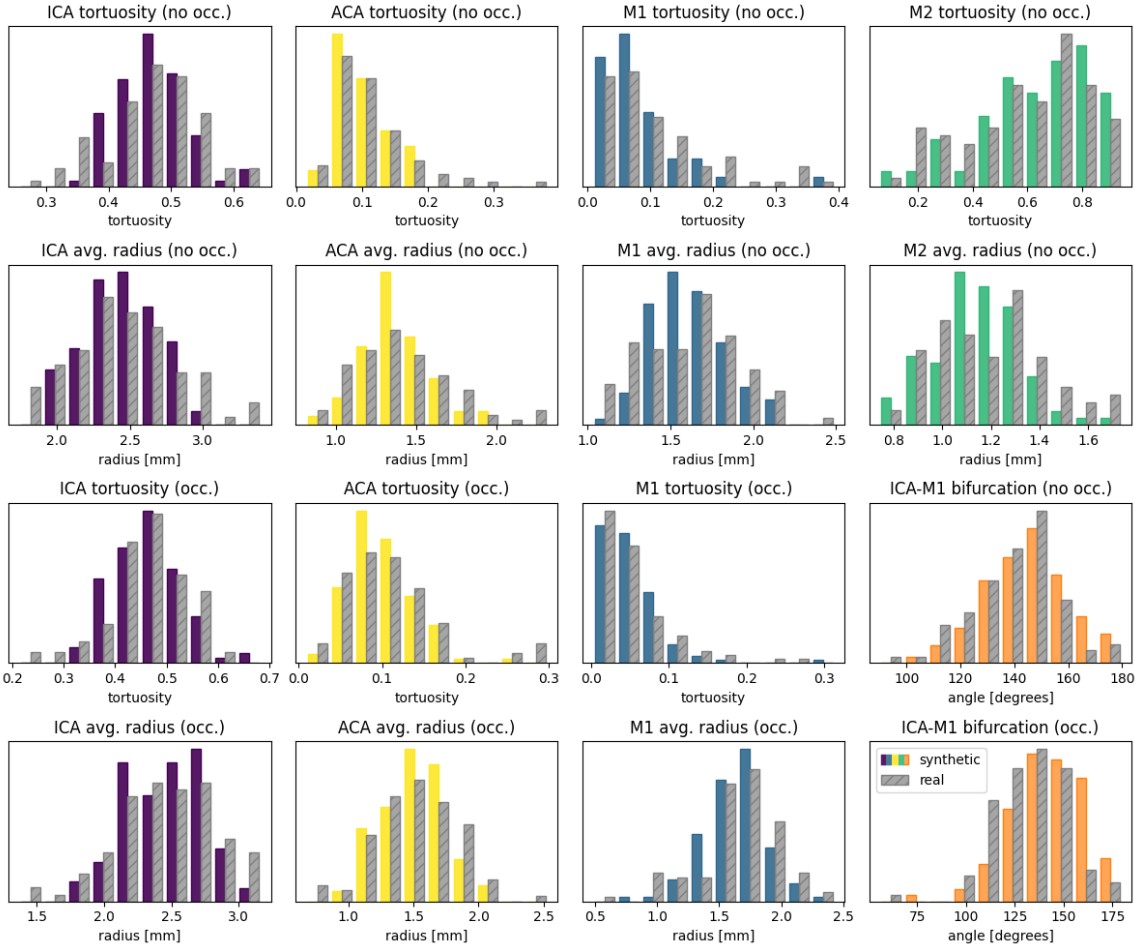

Figure 3: Distributions of tortuosity, average radius (in mm), and ICA-M1 bifurcation angle (in degrees) for the different vessel segments with and without an M1 occlusion.

## 3. Results and Evaluation

We train the diffusion model to generate cerebral vessel tree centerlines of 256 points, conditioned on an M1 occlusion. The training and sampling details are given in Section 3.1. In Section 3.2, we assess the geometric characteristics of the synthesized vessels, as well as model diversity. We quantitatively analyze the distributions of the geometric characteristics in Section 3.3.

### 3.1. Evaluation Methodology

We use the default hyperparameters from EDM (Karras et al., 2022) for the diffusion process, as tuning these did yield better results. The transformer architecture consists of 6 cross-attention blocks with 8 attention heads and a hidden dimension of 64 trained on centerlines. The model is trained for 2500 epochs with a batch size of 16 using the AdamW

(Loshchilov and Hutter, 2017) optimizer. We use a cosine annealing learning rate schedule, with an initial learning rate of $1e^{-7}$ that increases to $1e^{-4}$ in the first 100 epochs, after which it decreases back to $1e^{-7}$. We found these hyperparameters to result in accurate generative results while preventing the model from overfitting, which we validated on the held-out test set.

For evaluating the generative model, we sample a synthetic population of 200 samples, 100 of which contain an M1 occlusion. All synthetic samples are processed by the sequencing algorithm. Of the 200 samples, the sequencing algorithm failed on 7 M2 segments due to too much noise. We attribute this due to the large variance in M2 geometry combined with the small training set. For the sequencing algorithm, we set $\alpha = 0.25$ and $k = 5$. With these parameters, the algorithm successfully processes the train and test sets.

### 3.2. Qualitative Results and Diversity

The quality of the synthetic vessels is assessed by comparing the distributions of tortuosity, average radius of the ICA, ACA, M1, and M2 and ICA-M1 bifurcation angle between the synthetic data and the real training population in Figure 3. We observe that the distributions are specific to each vessel type and the presence of an M1-occlusion. Furthermore, the synthesized vessels represent the distributions of the geometric characteristics of the training set. This suggests that the model successfully captures the geometry of the vessel trees. Noteworthy, the model does not assign the wrong vessel type to the centerline points, e.g., assigning type ICA to a point belonging to the M1. We also assess the diversity of the synthesized population compared to the training set. We analyze the diversity by finding the closest sample from the training set for each sample in the synthetic population using the Chamfer distance. We observe that the model generates both diverse samples and samples that more closely resemble the training set, see Appendix A. Examples of generated vessels are shown in Figure 2.

### 3.3. Quantitative Analysis of Geometric Characteristics

We further analyze the distributions from Figure 3 by measuring their difference using the Kolmogorov-Smirnov (KS) test. When we compare the top and bottom rows of Figure 4, we observe that the differences between distributions between the synthetic vessels and the training set are similar to the differences within the training set itself. We also observe that the distributions of the vessels in the training set are specific to each vessel type and the presence of an M1-occlusion. The same specific distributions are seen in the synthetic population, which aligns with our observations in Section 3.2. This indicates that the distributions generated by the model are not arbitrary, but rather that the model successfully captures the specific geometries and conditioning from the training set.

## 4. Discussion and Conclusion

In this work, we presented a method for conditionally generating cerebral vessel trees with diffusion and a set-transformer backbone architecture. Our model generates vessel tree geometry parameterized by centerline points with associated radius and vessel type. Our model can generate complex and diverse cerebral vessel tree centerlines. In addition, the

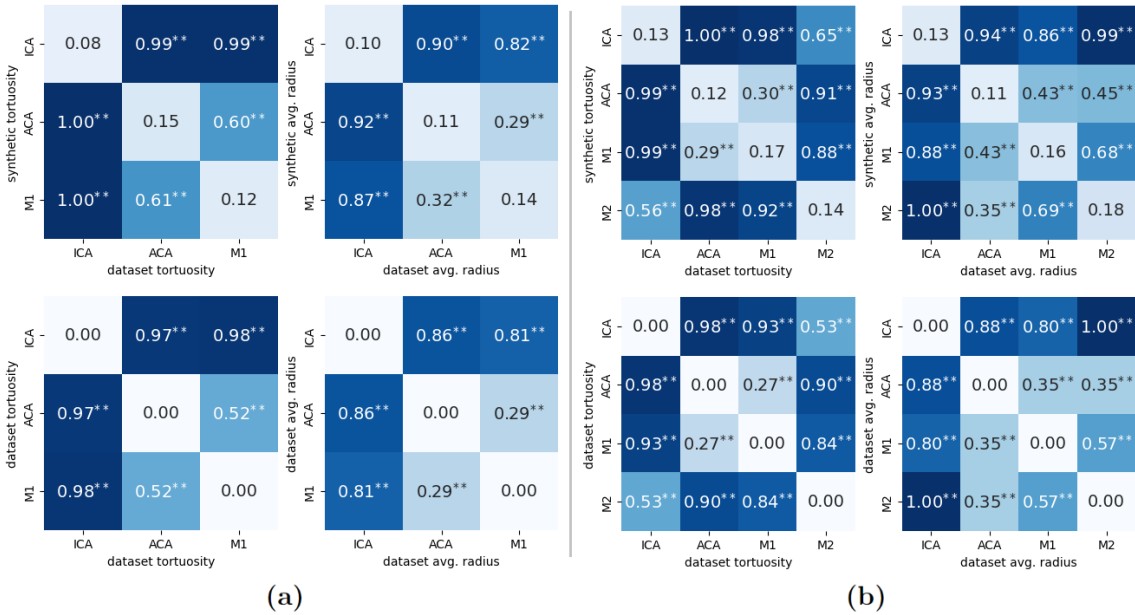

Figure 4: Differences between geometric characteristic distributions for samples with (a) and without (b) an M1-occlusion. *p-value < 0.05. **p-value < 0.01.

model labels each point with its corresponding vessel type. Furthermore, our simple sequencing algorithm is effective at connecting the individual points and vessel segments. Since the sequencing algorithm assumes equidistantly spaced points, this suggests the generated centerlines are relatively artifact-free.

Our experiments showed that our model can generate complex vessel trees that share the geometric characteristics of the training set. We observed that the model accurately captures the geometric differences between the individual vessel segments. The model can be effectively conditioned on auxiliary variables that affect vessel geometry, which we tested with the presence of an occlusion in the M1 vessel. Moreover, the model's generations are diverse and do not simply mimic the geometry from the training set. This suggests our model has learned an accurate distribution of cerebral vessel geometry.

It is important to consider that our evaluation does not directly validate the usefulness of our model on downstream tasks. However, centerline-based in-silico stroke treatment models have been developed and validated using the same dataset as our study (Luraghi et al., 2021; Miller et al., 2023). Since our model generates samples that are comparable to the real patient data, we are confident of the usefulness of our model for such computational models. In future studies, we will experiment on the required resolution and variation of the generated vessel trees to validate their usefulness for downstream tasks.

In conclusion, we showed that diffusion with a set-transformer architecture is a capable solution for conditionally generating realistic and diverse cerebral vessel tree geometry.

## Acknowledgment

This research was funded by the University of Amsterdam Responsible Digital Transformations (RDT) seed grant. Thijs P. Kuipers is an employee at inSteps B.V.; and is funded by AMC Medical Research (project number: 25158) and inSteps B.V.; Praneeta R. Konduri is a co-founder and shareholder of inSteps B.V.; and is funded by GEMINI (www.dth-gemini.eu): a European Union's Horizon research and innovation program (Grant Agreement Number: 101136438). The MR CLEAN Registry was partly funded by TWIN Foundation, Erasmus MC University Medical Center, Maastricht University Medical Center, and Amsterdam UMC.

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

## Appendix A. Chamfer Distance Distributions

The Chamfer distance can be used to calcualte the distance between two sets of points, e.g., point clouds. Given two sets of points $X$ and $Y$, the Chamfer distance $CD(X, Y)$ is:

$$CD(X,Y) = \frac{1}{|X|} \sum_{\mathbf{x}\in X} \min_{\mathbf{y}\in Y} ||\mathbf{x} - \mathbf{y}|| + \frac{1}{|Y|} \sum_{\mathbf{y}\in Y} \min_{\mathbf{x}\in X} ||\mathbf{y} - \mathbf{x}||, \tag{5}$$

where $|\cdot|$ denotes set cardinality. We determine the uniqueness of the generative model by calculating the Chamfer distance between each generated sample and the full training set. We consider the training sample that minimizes the Chamfer distance to be the sample that most closely resembles the generated sample. In figure 5, we report the minimum Chamfer distance between the generated samples and the training set.

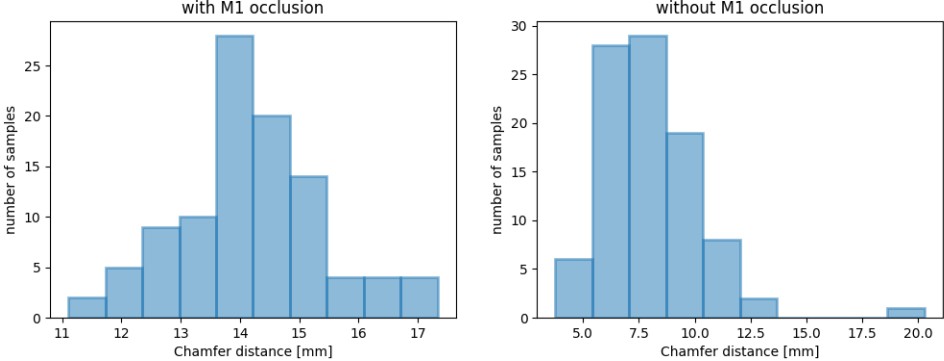

Figure 5: Minimum Chamfer distances between the generated samples and the training set for the samples with (left) and without (right) an M1 occlusion.

