# OpenReview forum: "Generating Cerebral Vessel Trees of Acute Ischemic Stroke Patients using Conditional Set-Diffusion"
_MIDL.io/2024/Conference — MIDL 2024 Oral_

### Official Review · Reviewer_bt6k · 2024-02-27

**Confidence:** 4
**Preliminary Rating:** 4
**Final Rating:** 4

**Summary:**

The authors propose a DDPM-based approach for generating cerebral vessel trees. Vessel centerlines are generated in a denoising process, using a transformer as the backbone network, with some post-processing steps to produce 'joined-up' trees. The model is conditioned on the presence of M1 occlusion. The authors assess generated samples by comparing their geometric characteristics, such as bifurcation angles, to those found in real data.

**Strengths:**

The paper is clear and well-written and seems to be a novel approach to generating new vessel trees. The method does seem to generate samples with similar geometric characteristics to the real data, without just memorising the training data.

The authors say they will make the code available.

**Weaknesses:**

My primary concern is the question of how good these generated trees would need to be to be useful in the in-silico clinical trials that motivate the work. While the authors show the generated trees follow similar distributions to those of real trees, we don't know how similar they need to be to be useful. It would have been nice to have some downstream validation of the generated trees on some sort of task to show these generated trees are 'good enough'.

**Detailed Comments:**

As far as I'm aware it is typical to standarise inputs to DDPMs so they have a variance of 1, not 0.5, as claimed in the paper. (How important the difference is in practice is another question).

Equation 1 suggests that noise is entirely additive. Generally in DDPMs the input is multiplied by a factor that is dependent on the timestep t, and also has a random noise component added. Can the authors clarify if their setup does differ from the standard setup?

It is unclear to me if the set transformer is a new contribution by the authors or an existing architecture. As far as I can tell it is a standard transformer with no positional embedding.

It might actually be useful to include positional embedding in the transformer on the sequence of coordinates. This might allow the model to generate ordered sequences centreline points and reduce the need for the sequencing post-processing step. Have the authors tried this approach?

**Justification Of Final Rating:**

I'm happy in general with the authors' answers and will keep my rating and weak accept. The authors were not able to address my questions about downstream validation of the usefulness of the generated trees, but I accept that is probably out of the scope of a conference paper with limited space, and think this paper is good overall.

**Justification Of The Preliminary Rating:**

A novel approach to the problem that seems to work well, clearly written, and a promise to make the code available. I think it would be strengthened by further validation of the samples, but I think this will be of interest to the community.

**Questions To Address In The Rebuttal:**

Is there any way the authors can validate the downstream usefulness of these generated trees?

I would like some responses to the questions about setup and architectures raised above, too.

---

> ### Author Response · Authors · 2024-03-17
> **Rebuttal - addressing individual reviewer's feedback**
>
> **Questions To Address In The Rebuttal:**
>
> * *“Is there any way the authors can validate the downstream usefulness of these generated trees?”*
>
>   **ANSWER:** We thank the reviewer for their feedback. Our proof-concept study aimed to evaluate the use of conditional set-diffusion for conditionally generating realistic vessel centerline data. The centerline data used in this study has been used to develop and validate in-silico stroke treatment models [Luraghi et al, 2021; Miller et al, 2023]. We are confident that our generated vessel centerlines can be utilized for such downstream tasks, since our model has proven to generate realistic synthetic data that is comparable to real patient data. Nevertheless, further experiments are required to learn about the required resolution and variation of our model  to generalize for downstream tasks. However, given the proof-of-concept nature of this study, performing such experiments remains outside of the scope of this study. We have now added these limitations and our intended future plans to perform the mentioned experiments  to the discussion section of our manuscript.
>
> **Detailed Comments:**
>
> * *“As far as I am aware, it is typical to standardize the input to DDPMs so they have a variance of 1, not 0.5, as claimed in the paper. (How important the difference is in practice is another question)”.*
>
>   **ANSWER:** We thank the reviewer for their comment. In our study, we use the default settings as formulated in the EDM [Karras, et al.] paper for our diffusion model. Hence, we set the std to 0.5. This std refers to the std of the target dataset, which does not need to be 1. The initial distribution from which noise is sampled does have an std of 1 (unit Gaussian). We now mention the 0.5 default from EDM in the method section 2.1.
>
> * *“Equation 1 suggests that noise is entirely additive. Generally in DDPMs the input is multiplied by a factor that is dependent on the timestep t, and also has a random noise component added. Can the authors clarify if their setup differs from the standard setup?”*
>
>   **ANSWER:** In the EDM [Karras, et al.] formulations, noise is indeed additive to the inputs. the noise is sampled from the Gaussian distribution N(0, sigma^2 * I), where sigma is the current noise level. Here, noise level can be interpreted as the time step from the general DDPM formulation. Note that some additional operations are performed after the noise has been added to the input, for example to take into account the standard deviation of the target (data) distribution (0.5 in our case). However, to keep the text accessible, we opted to not include these formulations in the current manuscript and rather reference the EDM paper that presents the full diffusion formulation. We added a more explicit reference to EDM for the diffusion hyperparameter setup in Results section 3.1.
>
> * *“It is unclear to me if the set transformer is a new contribution by the authors or an existing architecture. As far as I can tell it is a standard transformer with no positional embedding.”*
>
>   **ANSWER:** The set transformer is indeed a transformer without positional embeddings and it is not a new contribution. Instead, the novelty of our method stems from utilizing the permutation equivariance and unordered point sets for topology-free vessel tree generation. We highlighted this in the introduction section of our manuscript.
>
> * *“It might actually be useful to include positional embedding in the transformer on the sequence of coordinates. This might allow the model to generate ordered sequences' centreline points and reduce the need for the sequencing post-processing step. Have the authors tried this approach?”*
>
>   **ANSWER:** Ideally, we would like to have the model output an ordered set of points, so no post-processing would be required. One of the main design goals of our model was to be general, i.e., the model should be able to handle centerline trees with arbitrary topology, e.g., arbitrary number of branches, number of points per vessel segment, etc. However, such topology-free trees have no defined ordering of points, and treating them as a point cloud, i.e., a permutation equivariant fully connected graph, seemed more natural. Besides, under the assumption that the model generates well-defined lines with equidistantly spaced points, sequencing these lines is quite a straightforward task. We also hypothesize that making the model free of topological restrictions would allow it to better generalize to the geometry of the vessel tree. Nonetheless, in future studies, it is definitely interesting to experiment with having the model generate an ordered output, perhaps using a learned permutation matrix. We added our motivations for using unordered point sets in the introduction section.

---

### Official Review · Reviewer_uaH9 · 2024-02-27

**Confidence:** 2
**Preliminary Rating:** 4
**Recommendation:** Poster
**Final Rating:** 4

**Summary:**

This paper proposes to use a diffusion-based generative model and set transformers to generate a population of synthetic intracranial vessel tree centerlines with associated radii and vessel types. The model is conditioned on the presence of an occlusion in the middle cerebral artery, a frequently occurring location of occlusion in patients with Acute Ischemic Stroke (AIS). In addition to the set diffusion based generative model, the paper also uses a centerline sequencing technique to re-order the produced segments.

They evaluate the synthetically generated data against real data from the MRClean Registry (used during training) by comparing derivates of centerline co-ordinates and features. By examining the geometric properties of the generate vessels, they show that their model can accurately generate distributions of realistic looking cerebral vessel trees and reflect the geometric characteristics within real populations.

**Strengths:**

The methodological contribution of the work, i.e. the diffusion process and set transformers to generate realistic samples in conjunction with the reordering using centerline sequencing, is very interesting and possibly a novel way to use deep learning for this application. The paper is also well written and relatively easy and straightforward to follow. The experimental results are very reasonable and support the main claims.

**Weaknesses:**

1. It is not clear how the hyperparameters for the model are set in Section 3.1 since no validation data is mentioned. Similarly, how are the parameters for the diffusion model/set transformer determined?

2. While the analysis presented supports the claim that the model generates realistic looking samples (that match distributional characteristics for the training set), it is not clear whether the model generalizes in the sense of use for downstream applications. Please elaborate.

3. There are no comparisons performed against methods that currently exist in literature for this problem, more discussion here would be welcome.

**Detailed Comments:**

Additional questions:

1. How is the directional flow estimate obtained in Eq. 4?

2. Have the authors performed any analysis on the failure modes of the model leading to the non-convergence in the M1 occlusion cases mentioned in Section 3.1?

3. As mentioned in the discussion, how do you qualify whether the generated centerlines are free of artifacts?

**Justification Of Final Rating:**

I am satisfied with the author's rebuttal and it addresses the points I raised in the review adequately. This is an interesting proof of concept application and I would support the acceptance of the paper into the MIDL program.

**Justification Of The Preliminary Rating:**

I am leaning towards an accept based on the methodological novelty of the work for this application. However, there are some points that need additional clarification before publication, particularly on the utility of the framework against existing methods.

**Questions To Address In The Rebuttal:**

Please refer to weaknesses, particularly the points on the potential use-cases where the model may find appropriate use. Please also provide additional discussion on how the model compares to existing techniques used to address this problem.

**Special Issue:**

No

---

> ### Author Response · Authors · 2024-03-17
> **Rebuttal - addressing individual reviewer's feedback**
>
> **Questions To Address In The Rebuttal:**
>
> * *“It is not clear how the hyperparameters for the model are set in Section 3.1, ... . Similarly, how are the parameters for the diffusion model/set transformer determined?”*
>
>   **ANSWER**: We thank the reviewer for their feedback and we agree that a description of hyperparameter choices was lacking. Since we found that tuning the hyperparameters of the diffusion formulation did not improve our results, we set all diffusion hyperparameters to the default from the EDM [Karras et al, 2023] paper. We found that lowering the number of attention heads from 8 and the feature dimension from 64 drastically impacted the performance. Increasing these did not increase performance, but did result in overfitting. We added the motivations for these choices in the Results section 3.1.
>
> * *“... it is not clear whether the model generalizes in the sense of use for downstream applications. Please elaborate.”*
>
>   **ANSWER**: We thank the reviewer for their comment. In this study, we aimed to evaluate the use of conditional diffusion set models to generate realistic and conditioned vessel centerline data. Notably, the centerline data used in this study has been used to develop and validate computational stroke treatment simulation models [Luraghi et al, 2021, Miller et al, 2023]. Since our generated data has proven realistic and comparable with patient data, we are confident that vessel centerlines generated by our model can be utilized in the downstream tasks. Nevertheless, further experiments are necessary to learn about the required resolution and variation to generalize our model for such downstream tasks. However, given the proof-of-concept nature of this study to generate conditional vessel centerline data, performing such experiments remains outside the scope of this study. We have now added these limitations and future plans to the discussion section of our manuscript.
>
> * *“There are no comparisons performed against methods that currently exist in literature for this problem, more discussion here would be welcome.”*
>
>   **ANSWER**: To the best of our knowledge, we are the first to generate vessel centerline trees with arbitrary topology. Prior work either generates vessels without bifurcation (fixed topology), or cannot be conditioned on auxiliary variables. On the other hand, our qualitative and quantitative evaluation showed that the vessel centerlines generated from our model match those from the training set, while being unique implying that our model does not simply copy from the training set.
>
> **Detailed Comments:**
>
> * *“How is the directional flow estimate obtained in Eq. 4?”*
>
>   **ANSWER**: d_cur refers to the direction the centerline is currently moving in, and is calculated as the normalized difference between the point prior to the current last point, and the last point itself in the sequence.  The directions d_i from the last point in the sequence to the candidate points, are calculated as the normalized difference between candidate points and the last point added to the sequence. During sequence initialization, the sequence consists of only 1 point, so no directional information is used to determine the next point. We added the formulas for the directions to the revised manuscript.
>
>
> * *“Have the authors performed any analysis on the failure modes of the model leading to non-convergence in the M1 occlusion cases ...”*
>
>   **ANSWER**: We have indeed analyzed the failure cases of our model. All failure cases occurred  in the case where no M1 occlusion was present, specifically, in the synthesis of the M2 vessel.  A failure occurs when the post processing algorithm either cannot connect the individual points or when the vessel segment cannot be connected to the M1 segment. This could be explained by the inconsistencies in the M2 ground truth, especially in defining the end of the vessel segment, due to the inherent variation in the anatomy of M2, and the small training set. This could cause the M2 vessel to sometimes not converge, resulting in a noisy centerline which cannot be post-processed. We added our analysis to the Results 3.1.
>
> * *“As mentioned in the discussion, how do you qualify whether the generated centerlines are free of artifacts?”*
>
>   **ANSWER**: We qualify the synthetic sample as valid, or artifact-free, if the sequencing algorithm is able to reconstruct the centerline ordering, and the geometric characteristics of this ordering fall within the distribution of the training set. Artifacts generally present themselves in two ways: non-convergence of the generated centerline (noisy centerlines) or gaps in the generated centerlines. Since the post-processing step assumes points to be distributed equidistantly along the centerline, noisy distributed points and gaps in the centerline result in the failure of the post-processing step to form a connected vessel centerline tree. We revise this statement to highlight these details in the revised manuscript.

---

> ### Comment · Reviewer_uaH9 · 2024-03-26
> **Response to Rebuttal**
>
> I would like to thank the authors for taking the time to read through my comments and address them. I appreciate their effort in incorporating additional explanation to support the contribution and have no further concerns. Since I was leaning towards accepting the paper before the rebuttal, I stand by this decision.

---

### Official Review · Reviewer_iwJM · 2024-02-28

**Confidence:** 2
**Preliminary Rating:** 4
**Recommendation:** Oral
**Final Rating:** 4

**Summary:**

This article addresses the problem of generating realistic synthetic intracranial vascular centerlines, with or without occlusion in the M1 artery. The authors propose addressing this problem using a diffusion model with a set transformer, which takes as input the set of points of the intracranial centerlines (including radius and vessel branch information) while enforcing permutation invariance. The diffusion model can be conditioned by the presence or absence of an occlusion in the M1 artery.
Sampling in the diffusion model yields a set of centerline points without any relation between each other. The authors propose a post-processing algorithm to recover topology information by linking the points together and finding the corresponding bifurcation points.

They trained and evaluated their model on a private dataset of 110 patients. They validated their approach by comparing the distribution of geometric properties of their simulated vascular trees (radius and tortuosity of specific vessels) with the same distributions of the training set, with and without occlusion. They claim that their generated vascular trees share the same properties per vessel branch as the training set and that the conditioning affects the vascular properties as expected.

**Strengths:**

- The generation of realistic vascular typologies is of great interest to the community, and the ability to condition this generation with external information, such as the presence of an occlusion, is highly intriguing. This work lays the groundwork for more complex conditioning in vascular generation, which will undoubtedly attract considerable interest.
- The presented work demonstrates a high degree of novelty, leveraging state-of-the-art concepts to tackle the problem at hand.

**Weaknesses:**

- The description of the proposed approach may lack clarity and accessibility for individuals familiar with transformers and diffusion models but not experts in the most recent developments.
- The authors generate sets of points instead of centerlines, necessitating post-processing to recover the graph-like structure. The positioning of this work in relation to the graph diffusion model literature, which would seem more natural, is absent.
- The article lacks a summary figure describing the complete proposed approach.
- The generated vascular tree only contains a few vessels, and it is not clear whether this is a limitation of the proposed approach or done intentionally.

**Detailed Comments:**

Unordered centerline sequencing:
- How are d_cur and d_i calculated ?
- With the proposed segment merging approach, it seems that more than two segments can be merged. Is this the case? If so, is this a desired behavior, and does it happen in practice ?
- The authors should briefly explain why the sequencing algorithm failed on the seven samples.

The authors stated, "Noteworthy, the model never incorrectly classifies a vessel segment." I do not understand where in the pipeline this approach classifies the vessel type. It seems that a point has a vessel type, and the sequencing algorithm links points of the same type. The authors should provide more details on this sentence.

**Justification Of Final Rating:**

I thank the authors for having carefully reply to my questions and comments.
The description of the approach seems clearer now.
I still have doubts regarding the choice of the point cloud instead of a graph, particularly concerning the scalability and complexity of generating larger vascular trees. However, the authors have provided adequate justification for their choice, which satisfies my initial question. Consequently, I maintain my original rating and believe that the article is suitable for publication in MIDL.

**Justification Of The Preliminary Rating:**

This article tackles a very interesting subject and propose an innovative method to generate realistic vascular topologies. Nevertheless, the methodology could benefit from a clearer description to facilitate understanding and further development by the community.

**Questions To Address In The Rebuttal:**

- The input of the approach consists of centerlines, which could be more appropriately represented by a graph rather than a set of points. The authors should provide commentary on their decision to use the set of point representation instead of employing a graph diffusion model.
- It remains unclear how the set transformer and the diffusion process interact. Is the set transformer the backbone architecture of the diffusion model ?  It seems that it is not the case, as Figure 1 outputs the predicted centerlines instead of the predicted noise. Alternatively, does the diffusion model have a classic UNet backbone, and the set transformer is trained separately by sampling in the diffusion model to perform denoising ? The authors should clarify this aspect and add an overview figure of their approach.
- The authors should explicitly state their contributions in the introduction section.
- Quantitative results on the distribution of Chamfer distances should be added to the article to support the observations made in section 3.2. These results could be included in the appendix.

---

> ### Author Response · Authors · 2024-03-17
> **Rebuttal - addressing individual reviewer's feedback.**
>
> **Questions To Address Rebuttal:**
>
> * *“... The authors should provide commentary on their decision to use the set of point representation instead of employing a graph diffusion model?”*.
>
>     **ANSWER:** We thank the reviewer for their question. By representing the vessel trees as point clouds, we essentially treat them as a fully connected graph through the message passing framework in the self-attention mechanism. Unlike graph representations that limit the model to only the neighboring points, the point cloud representation enables the model to learn about the geometry on a global scale and makes our model applicable to trees with arbitrary topology. We highlighted our motivations for using point sets in the introduction.
>
> * *“It remains unclear how the set transformer and the diffusion process interact. … The authors should clarify this aspect and add an overview figure of their approach.”*
>
>     **ANSWER:** Thank you for addressing these issues. The transformer model is indeed the backbone of the diffusion process, modeling the denoising function.. The forward diffusion process adds noise to the point cloud, and the transformer is tasked to denoise the point cloud. During sampling, the initial noise is sampled from the unit Gaussian and iteratively denoised to form the centerline point cloud that includes. We apply the centerline sequencing post-processing step to obtain the final ordered vessel tree. This has been clarified in the method section, as well as in the new overview Figure 1.
>
> * *“Missing contribution section in the introduction and the distributions of chamfer distances.”*
>
>     **ANSWER:** We added the contribution section to the introduction and the distributions of the chamfer distances to the appendix section A.
>
> **Weaknesses**:
>
> For the first two points, please refer to the answers for the rebuttal questions above.
>
> * *“The article lacks a summary figure describing the complete proposed approach.”*
>
>     **ANSWER:** In line with revising the methodology to make the description of the current method more clear, we have revised Figure 1 to have it represent an accurate representation of the proposed method.
>
> * *“The generated vascular tree only contains a few vessels, and it is not clear whether this is a limitation of the proposed approach or done intentionally.”*
>
>     **ANSWER:** We thank the reviewer for their remark. Large vessel occlusions most frequently occur in the arteries of the anterior circulation such as ICA, ACA, M1, and M2. To increase the usefulness of the centerline data generated by our model to be relevant for downstream tasks such as in computational models of stroke treatment simulation, we chose to focus on the few larger arteries. Nonetheless, since we consider vessel trees as point clouds, our model is not constrained by specific topology. We highlighted our choices in the introduction section.
>
> **Detailed Comments:**
>
> * *“How are d_cur and d_i calculated?”*
>
>     **ANSWER:** d_cur refers to the direction the centerline is currently moving in, and is calculated as the normalized difference between the point prior to the current last point, and the last point itself in the sequence.  The directions d_i from the last point in the sequence to the candidate points, are calculated as the normalized difference between candidate points and the last point added to the sequence. During sequence initialization, the sequence consists of only 1 point, so no directional information is used to determine the next point. We added the formulas for calculating the directions in the revised manuscript.
>
> * *“With the proposed segment merging approach, it seems that more than two segments can be merged. Is this the case? If so, is this behavior desired, and does it happen in practice?”*
>
>     **ANSWER:** It is indeed the case that more than two segments can be merged. This is intentional, since we wanted the sequencing algorithm to be general, meaning it should work with any tree topology – similarly to the diffusion model itself. It is, for example, possible for trees to have tri-furcations, or even ‘higher-furcations’. However, our data only contains bifurcations, and we did not see the merging algorithm merge more than two segments.
>
> * *“The authors have stated, “Noteworthy, the model never incorrectly classifies a vessel segment.” … The authors should provide more details on this sentence.”*
>
>     **ANSWER:** We thank the reviewer for pointing this out. Indeed, ‘classifying’ is an incorrect terminology. At the start of the reverse diffusion process, the xyz location, radius, and vessel type are unknown. Henc, during the reverse diffusion process, the model has to find the right xyz locations and radius for this point, as well as assign the right type to it. We observed that our model does not incorrectly assign a vessel type to a point, for example, assign type M1 to a point that has been placed in the ICA segment. We have clarified our statement in the results Section 3.2.

---

### Author Response · Authors · 2024-03-17
**Rebuttal**

We would like to thank the reviewers for their time and valuable feedback which has allowed us to improve our work. We are also happy to hear that our work was found to be of interest and  well-written. In the following paragraphs, we summarize the feedback of the reviewers along with our answers and changes to the manuscript. We then address the questions raised by each reviewer in detail. For all specific changes made to the manuscript to address the feedback of the reviewers, we refer to the revised manuscript, where all changes are highlighted in red.

All reviewers raised minor questions regarding the model architecture and study design choices, specifically regarding the interaction of the set-transformer and the diffusion process, the choice of points without positional embeddings instead of graphs, and the selection of hyperparameters. In essence, the diffusion process is tasked with denoising a set of randomly sampled points to form a centerline tree. The set-transformer performs this denoising. We made these details more explicit in the manuscript, and added these details to an overview figure (Figure 1). The centerline trees are represented by an unordered set of points, as this makes our approach topology-free, and learn about the global geometry. We added these motivations in our manuscript. We used the default diffusion parameters from EDM [Karras et al, 2022], as we found that tuning these parameters did not improve our results. We made our hyperparameter choices more explicit in the manuscript. The hyperparameters of the transformer were tuned to decrease model run-time, and prevent overfitting without impacting our results. This was tested on the held-out test set. We added our motivation for the chosen hyperparameters to the manuscript.

Reviewers uaH9 and bt7k asked about validating and the usefulness of the generated data for downstream tasks. Our model generates vessel centerline data that can serve as input to computational treatment simulations, such as the virtual treatment procedure proposed by [Luraghi et al, 2021; Miller et al, 2023], which was developed using the same dataset as our study. Since our synthetic data generated by our model is proven to be comparable to real-world patients, we are confident that our model is useful for such computational treatment models. Nonetheless, we agree that additional  experiments on the resolution and variation of the synthetic data are necessary to validate the usefulness for downstream tasks. However, given the proof-of-concept nature of this study, we are unfortunately unable to perform such experiments. We have therefore added these limitations and future plans to the discussion of our manuscript.

Lastly, reviewers uaH9 and iwJM highlighted missing details from our manuscript on comparisons with literature, a contribution section in our Introduction, and missing Chamfer distance distributions to determine uniqueness quality of our model. To the best of our knowledge, we are the first to introduce a conditional topology free centerline vessel tree generative method. We do note that our quantitative and qualitative analysis are in line with existing work [Bridio et al, 2023; Wolterink et al, 2018] on generating vessel centerlines. If the reviewer has any particular existing work in mind, we would gladly incorporate it into the manuscript. We have now incorporated the missing contribution section in the introduction and added an Appendix section where we present the Chamfer distance distributions.

**References:**

* Giulia Luraghi, Sara Bridio, Jose Felix Rodriguez Matas, Gabriele Dubini, Nikki Boodt, Frank JH Gijsen, Aad van der Lugt, Behrooz Fereidoonnezhad, Kevin M Moerman, Patrick McGarry, et al. The first virtual patient-specific thrombectomy procedure. Journal of Biomechanics, 126:110622, 2021.


* Miller, C., Konduri, P., Bridio, S., Luraghi, G., Terreros, N. A., Boodt, N., ... & Hoekstra, A. (2023). In silico thrombectomy trials for acute ischemic stroke. Computer Methods and Programs in Biomedicine, 228, 107244.

* Tero Karras, Miika Aittala, Timo Aila, and Samuli Laine. Elucidating the design space of diffusion-based generative models. Advances in Neural Information Processing Systems, 35:26565–26577, 2022.

* Jelmer M Wolterink, Tim Leiner, and Ivana Isgum. Blood vessel geometry synthesis using generative adversarial networks. arXiv preprint arXiv:1804.04381, 2018.

* Sara Bridio, Giulia Luraghi, Anna Ramella, Jose Felix Rodriguez Matas, Gabriele Dubini, Claudio A Luisi, Michael Neidlin, Praneeta Konduri, Nerea Arrarte Terreros, Henk A Marquering, et al. Generation of a virtual cohort of patients for in silico trials of acute ischemic stroke treatments. Applied Sciences, 13(18):10074, 2023.

---

### Meta-Review · Area_Chair_yf32 · 2024-04-04

**Recommendation:** Accept (Oral)
**Confidence:** 4

**Metareview:**

Reviewers appreciated both the methodological originality of the approach as well as the quality of the generation of vasculature itself which seems realistic. The reviewers also appreciated the clarity of the paper.

---

### Decision · Program_Chairs · 2024-04-06

Accept (Oral)